# Serum Metabolomic Profiling in Aging Mice Using Liquid Chromatography—Mass Spectrometry

**DOI:** 10.3390/biom12111594

**Published:** 2022-10-29

**Authors:** Tong Yue, Huiling Tan, Yu Shi, Mengyun Xu, Sihui Luo, Jianping Weng, Suowen Xu

**Affiliations:** Department of Endocrinology, Institute of Endocrine and Metabolic Diseases, The First Affiliated Hospital of USTC, Division of Life Sciences and Medicine, Clinical Research Hospital of Chinese Academy of Sciences (Hefei), University of Science and Technology of China, Hefei 230001, China

**Keywords:** aging, metabolomics, biomarkers, machine learning

## Abstract

Background: The process of aging and metabolism are intricately linked, thus rendering the identification of reliable biomarkers related to metabolism crucial for delaying the aging process. However, research of reliable markers that reflect aging profiles based on machine learning is scarce. Methods: Serum samples were obtained from aged mice (18-month-old) and young mice (3-month-old). LC-MS was used to perform a comprehensive analysis of the serum metabolome and machine learning was used to screen potential aging-related biomarkers. Results: In total, aging mice were characterized by 54 different metabolites when compared to control mice with criteria: VIP ≥ 1, *q*-value < 0.05, and Fold-Change ≥ 1.2 or ≤0.83. These metabolites were mostly involved in fatty acid biosynthesis, cysteine and methionine metabolism, D-glutamine and D-glutamate metabolism, and the citrate cycle (TCA cycle). We merged the comprehensive analysis and four algorithms (LR, GNB, SVM, and RF) to screen aging-related biomarkers, leading to the recognition of oleic acid. In addition, five metabolites were identified as novel aging-related indicators, including oleic acid, citric acid, D-glutamine, trypophol, and L-methionine. Conclusions: Changes in the metabolism of fatty acids and conjugates, organic acids, and amino acids were identified as metabolic dysregulation related to aging. This study revealed the metabolic profile of aging and provided insights into novel potential therapeutic targets for delaying the effects of aging.

## 1. Introduction

Aging is a time-dependent process linked to various diseases, including cardiovascular disease, neurodegenerative disease, diabetes, and cancer [1]. As the global aging population grows progressively, the social, economic, and healthcare burdens we face as a society are steadily increasing. It is well known that the pathophysiology and mechanisms of aging are complex, and aging-related mechanisms involve mitochondrial dysfunction, immune dysfunction, autophagy alterations, and other complex factors [2,3,4]. Therefore, there is an unmet medical need to detect early and develop novel interventional strategies to address aging and age-related diseases.

Nonetheless, aging is a highly heterogeneous process that is difficult to characterize precisely. The multi-omics technique, including genomics, transcriptomics, metabolomics, and proteomics, has been applied in recent aging-related studies [5]. Since aging and age-related disease studies almost invariably point to dynamic alterations in the metabolome, metabolomics technologies have been the central theme of high-throughput omics technologies in aging investigations [6,7,8]. As an emerging omics technology, metabolomics can monitor the profile and alterations of endogenous small-molecule metabolites in tissues and has been gradually used in recent decades to screen biomarkers linked to metabolic diseases and help clarify relevant biochemical pathways [9]. Moreover, current studies suggest that though the metabolite profile was captured ten years ago and that the dynamic concentration changes of metabolites can predict the development of some aging-related diseases such as diabetes, cardiovascular disease, and age-related cognitive decline [10,11,12]. In addition to the predicted function of aging-related diseases, metabolomics has been used to investigate death risk biomarkers. For instance, 14 circulating biomarkers were independently related to all-cause mortality in a recent analysis of large populations [13]. The metabolomics analysis in the blood of 15 young and 15 elderly people showed that blood metabolites are highly valuable for human aging research [14]. These metabolites were considered promising and convenient approaches to monitoring aging and related diseases.

Considering the technical difficulties in obtaining blood metabolite cellular components that are labile, serum metabolomics has been well-investigated to explore disease biomarkers and understand disease mechanisms. Furthermore, disease mechanisms can differ among individuals, necessitating precision medicine in the future [15]. To achieve this aim, serum metabolomics, a high-throughput technology, which can measure thousands of compounds simultaneously, is becoming an essential technology for aging and related disease studies. A previous study showed that several metabolites could be considered general age markers using different machine learning algorithms [16]. Mice have been used as a typical organism model in serum metabolomics to discover the biomarkers and metabolic pathways that regulate lifespan. In this study, we aimed to investigate the plasma metabolome profile of aging mice in comparison with that of young mice.

## 2. Materials and Methods

### 2.1. Mice Husbandry and Serum Preparation

We used eight male aged C57BL/6J mice (18 months) and eight young control mice (3 months) purchased from Hangzhou Ziyuan Laboratory Animal Technology Co., Ltd. (Hangzhou, China). The aging mice and control mice were fed a chow diet (Appendix A). The caloric composition of the chow diet was 15.8% fat, 20.3% protein, and 63.9% carbohydrate. The Animal Care and Use Committee of the University of Science and Technology of China (USTC) authorized all animal experiments. At the experimental endpoint, blood was collected retro-orbitally. Blood was centrifuged at 3000× *g* for 10 min (4 °C) to collect serum. The serum was stored at −80 °C for further metabolome analysis. 

### 2.2. Sample Preparation

Metabolite extraction mainly was carried out using the procedures described below. At 4 °C, frozen plasma was thawed and dissolved. Internal standards mix 1 (IS1) and internal standards mix 2 (IS2) were added to the 100 μL samples. The samples were extracted by immediately adding 300 L of precooled methanol and acetonitrile (2:1, *v*/*v*) to the 100 μL samples for quality control of the sample preparation. Samples were centrifuged for 20 min at 4000 rpm after being vortexed for 1 minute and incubated at −20 °C for 2 hours. After that, the supernatant was vacuum freeze dried. Metabolites were then resuspended in 50% methanol (150 µL) and centrifuged for 30 min at 4000 rpm before being transferred to autosampler vials for further analysis. The stability and repeatability of the metabolomics profiling system are essential for whole LC-MS analysis, and a quality control (QC) sample was established [17]. The samples were randomly sorted to decrease system errors and give more trustworthy experimental results. For every ten samples, a QC sample was interposed.

### 2.3. UPLC-Orbitrap MS Condition

The serum metabolites were analyzed on Waters 2D UPLC (Waters, Milford, MA, USA) using a Waters 2D UPLC (Waters, USA) linked to a Thermo Fisher Scientific Q-;Exactive mass spectrometer with a heated electrospray ionization (HESI) source and the Xcalibur 2.3 software application (Thermo Fisher Scientific, Waltham, MA, USA). Then, we used Waters ACQUITY UPLC BEH C18 column (1.7 μm, 2.1 mm × 100 mm, Waters, Milford, MA, USA) to complete the chromatographic separation and kept the column temperature constant at 45 °C. In the positive mode, the mobile phase was 0.1% formic acid (A) and acetonitrile (B). The mobile phase included 10 mM ammonium formate (A) and acetonitrile (B) in the negative mode. The mass spectrometric settings for the two ionization modes were given in standard methods [18]. The identification of metabolites meets the levels published by the Metabolomics Standards Initiative (MSI). [19]

### 2.4. Serum Metabolite Analysis

The LC-MS/MS technology was performed to profile serum metabolites. The high-resolution mass spectrometer Q Exactive (Thermo Fisher Scientific, USA) collected negative and positive ion data. The raw LC-MS data were processed with The Compound Discoverer 3.1 (Thermo Fisher Scientific, USA). The multivariate raw data was dimensionally reduced by PCA and OPLS-DA analyses. The criteria including the variable importance projection (VIP) value was used with the variability analysis, the Student’s *t*-test, and the fold change (VIP ≥ 1, *q*-value < 0.05, and Fold-Change ≥ 1.2 or ≤0.83) was performed to screen potential biomarkers. HMDB, KEGG, and mzCloud databases were examined to identify metabolites. Figure 1 summarizes the methodology for a comprehensive metabolomics investigation in aging mice.

### 2.5. Pathway Analysis

MetaboAnalyst 5.0 was used to perform pathway analysis on the significantly changed metabolites. To define the highly enriched pathways, the Benjamini–Hochberg approach was employed to calculate the adjusted *p*-value and *p*-value < 0.05 [20]. 

### 2.6. Statistical Analysis

Mann–Whitney U tests were used to compare the aging group with the control group. The metabolites and profile differences between the aging and control mice groups were determined using unsupervised principal component analysis (PCA) applied to common Pareto correction and supervised projections to latent structures-discriminant analysis (OPLS-DA). A 10-fold method with unit variance scaling was used to cross-validate the OPLS-DA models. The fitting condition for the OPLS-DA model was evaluated using the parameter R^2^, and the prediction ability was assessed using the parameter Q^2^. To screen the differential metabolites, the VIP (Variable Importance in Projection) values of the first two main components of the OPLS-DA model were coupled with the variability analysis, the Fold change, and the Student’s *t*-test. Four algorithms, Logistic Regression, Support Vector Machine, Gaussian Naive Bayes, and Random Forest were used for machine learning by Python 3. The four algorithms of machine learning use eight-cross validation to randomly divide the data set into training data and test data. Receiver operator characteristic (ROC) curve analysis was generated by Monte-Carlo cross-validation (MCCV) using balanced sub-sampling. Each MCCV uses two-thirds (2/3) of the samples to evaluate feature importance. On the 1/3 of samples that were left out, the top 2, 3, 5, 10… 100 (max) features were used to build classification models. Each model’s performance and confidence interval were calculated multiple times.

## 3. Results

### 3.1. PCA and OPLS-DA of Serum Samples in Aging Mouse Models

PCA and OPLS-DA were performed for both negative and positive ionization modes. We constructed these models to determine metabolic differences and confirm the LC-MS/MS system’s stability. Different distribution patterns of metabolites between the aging group and controls were observed, indicating the applicability of the two models and changes in serum metabolite composition in aging mice (Figure 2). As shown in Figure 2a,b, significant segregation is shown in the PCA score plot, indicating an overall change in the serum metabolite composition in aging mice. Our results indicated the OPLS-DA model’s high classification and prediction abilities. In negative ion mode, the R^2^Y and Q^2^ values were 0.858 and 0.753, respectively, whereas, in positive ion mode, they were 0.898 and 0.808. These results demonstrated that the models were able to differentiate between the two groups and identify discriminating metabolites.

### 3.2. Differential Metabolite Identification in Aging Mice

The cluster profile was capable of differentiating aged groups from the control group (Figure 3a,b). The VIP, *q*-values, and Fold-Change (VIP ≥ 1, *q*-value < 0.05, and Fold-Change ≥ 1.2 or ≤0.83) were used to screen the differential metabolites. In negative ion mode, there were 62 up-regulated and 119 down-regulated metabolites in aging mice (Figure 3c). In positive ion mode, there were 174 up-regulated and 291 down-regulated metabolites (Figure 3d). A total of 54 metabolites were identified in the HMDB database. We then set stricter selection criteria with *p* < 0.0001 and a *q*-value < 0.01, recognizing 20 metabolites (Table 1). Among these metabolites, aging mice had higher levels of fatty acids and conjugates (oleic acid and palmitoleic acid), organic acids (citric acid and alpha-ketoglutaric acid), and lower levels of amino acids (l-methionine and formyl-l-methionyl peptide) and indoles (tryptophol), among the total altered metabolites that were the most discriminative for classification between the two groups.

### 3.3. Pathway Enrichment Analysis

The pathways with significant differences in negative ion mode are (1) fatty acid biosynthesis; (2) phenylalanine metabolism; (3) cysteine and methionine metabolism; and (4) vascular smooth muscle contraction (Figure 4a). In positive ion mode, the top ten significant pathways are the (1) biosynthesis of amino acids; (2) caffeine metabolism; (3) cysteine and methionine metabolism; (4) linoleic acid metabolism; (5) metabolic pathways; (6) D-glutamine and D-glutamate metabolism; (7) alpha-Linolenic acid metabolism; (8) lysine degradation; (9) citrate cycle (TCA cycle); and (10) glyoxylate and dicarboxylate metabolism. The whole pathways and impact factors were shown in Appendix A (*p* < 0.05). Notably, the increased levels of fatty acids and conjugates in the aging mice could correlate well with the higher abundances of fatty acid biosynthesis metabolism. Moreover, alterations in amino acids and indoles metabolites often affect the functional and metabolic states; our data showed that amino acids and indoles are involved in cysteine and methionine metabolism and the TCA cycle, indicating their interactions were closely associated with the dysfunctional metabolism in the aging process.

### 3.4. Machine Learning for the Candidate Biomarkers

We used four machine learning algorithms (Logistic Regression, Support Vector Machine, Gaussian Naive Bayes, and Random Forest) because the data was characterized by limited sample sizes with high dimensionality. The mean accuracy was 0.9375 for Logistic Regression, Support Vector Machine, and Random Forest in negative ion; the mean accuracy was 0.9375 for Logistic Regression, Gaussian Naive Bayes, and Random Forest in positive ion. Next, we applied the Logistic Regression algorithm to select the top ten features in both modes. The candidate metabolites are shown in Appendix A. 

In order to better screen the strict metabolites, we merged the candidate biomarkers of machine learning and the VIP, *q*-values, and Fold-Change selection criteria, which recognized oleic acid. The ROC analysis showed that oleic acid presented the highest discrimination value (Figure 4c; AUC ≥ 0.95). In addition, four metabolites (*p* < 0.001) were selected as prospective biomarkers for aging, namely, citric acid, D-glutamine, trypophol, and L-methionine. The AUCs were 1.0, 0.875, 0.953, and 0.984, respectively (Figure 5). The AUCs for the five altered serum metabolites were over 0.95, indicating that these candidate metabolites perfectly separated the aging and the young groups (Appendix A). We applied these five potential metabolites to distinguish and predict the aging status.

## 4. Discussion

Aging is a natural physiological process that is accompanied by a wide spectrum of functional and metabolic degeneration. Despite considerable efforts and our advanced knowledge of aging pathologies, the underlying mechanism that causes aging is still largely unknown [21,22,23,24]. It is essential to discover novel biomarkers to understand the mechanism and develop novel therapies to delay aging. Several studies have looked for the metabolic factors that influence lifespan [25,26,27] and metabolomics has recently gained attention as a promising omics tool for identifying biomarkers and classifying disease [28]. In our study, we employed high-performance LC/MS-based metabolomic profiling to find differential serum metabolites in aging and control mice. 

Biomarkers of aging may be used to treat age-related disorders. Individuals’ physiological states and the underlying mechanisms connected to homeostatic changes during their lives are reflected by aging biomarkers [29]. Machine learning can improve diagnosis and treatment, transforming biomedicine and the medical practice [15]. In recent studies, machine learning has been used to discover genetic and metabolite markers linked to aging [30,31,32,33]. We selected 20 candidate metabolites related to aging using four algorithms, including LR, GNB, SVM, and RF. Age-related increases in saturated fatty acid levels, glutamine, and tryptophan are potential contributors to aging [34]. We also identified oleic acid, which was not previously associated with aging and offers a potential pathway that counteracts aging. Oleic acid is one of the mono-unsaturated fatty acids that affects cellular signaling and fatty acid biosynthesis pathways. One study has shown that as the rats aged, the absorption of oleic acid in the intestine of rats increased [35]. We found that other dietary fats such as myristic acid and palmitoleic acid were also increased in aging mice. The significance of oleic acid in decreasing insulin resistance and enhancing insulin transport in diabetics has been studied extensively [36,37]. However, previous studies showed that these dietary fat concentrations were not associated with heart failure (HF) or cardiovascular disease (CVD) risk [38,39]. 

Furthermore, pathway enrichment analysis also revealed a disordered cysteine and methionine metabolism pathway, and the levels of relevant metabolites were decreased. Cysteine and the methionine metabolism pathway support various metabolic processes, including nucleotide synthesis and redox homeostasis [40]. Zhu et al. demonstrated that trans-sulfuration-mediated cysteine synthesis is essential for tumor growth in vivo [41]. Consistent with a previous study [42], we found l-methionine, a peptide involved in cysteine and methionine metabolism, decreases with age in serum, indicating its potential anti-aging function. Methionine as an essential amino acid plays a crucial role in modulating health [43]. Recent studies suggest that methionine has been identified as a potential biomarker to extend lifespan, mediating various nutrient and genetic perturbations. Methionine restriction can delay aging and extend longevity [44]; Lees et al. reported that methionine-restricted feeding could reverse the adverse effects of aging on hepatic lipogenic gene expression, adiposity, body mass, and insulin resistance through fibroblast growth factor 21 signaling [45]. Methionine is also linked to the trans-sulfuration pathway (TSP), which results in higher hydrogen sulfide (H_2_S) synthesis and an increased lifespan of yeast [46]. Methionine restriction also emerges as a pro-angiogenic trigger, promoting VEGF expression/activity, motility, and sprouting in ECs in vitro, thus facilitating the outgrowth of new blood vessels from existing endothelial cells [47]. Our data indicate that different approaches, including cell growth, movement, and metabolism, resulted in diverse clinical outcomes in the aging process.

There was the deregulated metabolism of other metabolites related to aging including tryptophol and D-glutamine, which are the significant degradation metabolites of tryptophan and D-glutamine and D-glutamate metabolism. Glutamine is the most common amino acid in circulation and has been linked to the TCA cycle [48]. D-glutamic acid was first detected in mammalian tissues, and glutamine metabolism is associated with advanced age [49,50]. A study has shown that different metabolites present in 3-month-old and 24-month-old male mice are involved in D-glutamine and D-glutamate metabolism, which may contribute to changes in renal dysfunction [51]. In addition, individuals with age-related disorders, such as major depressive disorder (MDD), had a significantly higher ratio of glutamine to glutamate than normal older adults at baseline [52]. However, D-glutamine and D-glutamate metabolism occurs in bacteria [53]. It is plausible that altered gut microbiota composition, thus altering the percentage of D-glutamine and D-glutamate-producing bacteria, in particular, may lead to the deregulated metabolism of D-glutamine and D-glutamate in aging mouse serum. Tryptophol was involved in tryptophan metabolism and identified as a potential biomarker of depression [54]. Although, to the best of our knowledge, no research to date has investigated the impact of the age-related decline in tryptophol levels on aging. Overall, our data revealed that the serum metabolomic of the aging mice was remarkably altered, featuring higher levels of several fatty acids and organic acids and lower levels of amino acids and indoles. We speculate that further research is warranted to investigate the mechanism between tryptophol, D-glutamine, and lifespan. 

We have to recognize that our study had several limitations. First, although we used machine learning, the sample size was small, which may lead to unintentional bias. The five metabolites identified need to be validated in clinical cohorts; thus, our findings should be regarded as hypothesis-generating, implying that these biomarkers could be used in humans and that further research into these potential metabolites in human cohorts is needed.

In summary, we generated a metabolomic profile of serum comparing aging mice and control mice. We identified oleic acid, citric acid, D-glutamine, trypophol, and L-methionine to be correlated with aging based on different algorithms and applied machine learning. Our study also indicated that these five potential biomarkers can be used as biomarkers to predict aging, with an AUC greater than 0.85. Further research is needed to confirm these findings in humans and determine their longitudinal associations. 

## 5. Conclusions

We identified 54 aberrant serum metabolites in aging mice by a high-throughput LC/MS-based metabolomics investigation. According to the metabolic pathway analysis, these potential biomarkers were mostly involved in fatty acid biosynthesis, cysteine and methionine metabolism, D-glutamine and D-glutamate metabolism, and the citrate cycle (TCA cycle). Based on different algorithms and applied machine learning, five potential biomarkers showed a high diagnostic value with an AUC greater than 0.85. Furthermore, our findings provide comprehensive insights into the aging metabolism profile. These combined serum metabolites may be a predictive or practical technique for screening aging biomarkers and responding with therapeutic interventions.

## Figures and Tables

**Figure 1 biomolecules-12-01594-f001:**
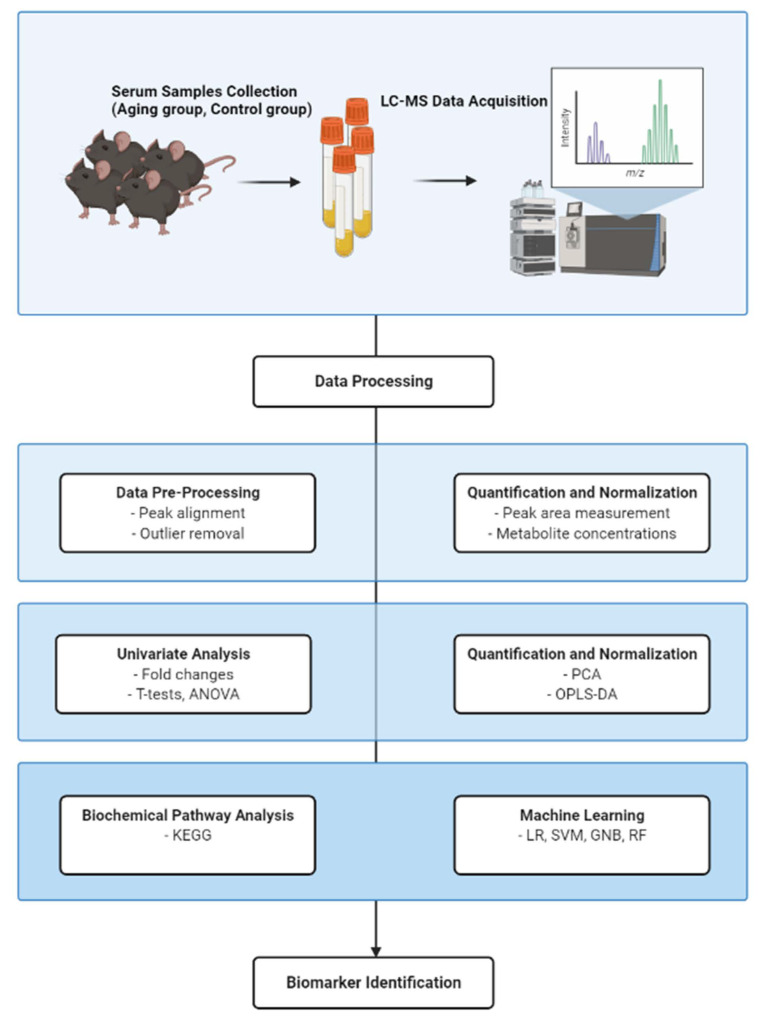
Overview of the workflow of the comprehensive analysis of metabolome in aging mice. (**a**): Serum samples were collected from aging mice and control mice, and the metabolite data were obtained by LC-MS/MS. (**b**,**c**): Data processing by peak extraction, peak alignment, and metabolite identification. (**d**): The results of Fold change and Student’s T-Test obtained from univariate analysis were combined to screen for differential metabolites. (**e**): PCA and OPLS-DA were used to reduce the dimensionality of multivariate raw data, and the similarity and difference between and within the sample groups were obtained. (**f**,**g**): Metabolite classification and functional annotation were carried out, and machine learning was used to search for biomarkers.

**Figure 2 biomolecules-12-01594-f002:**
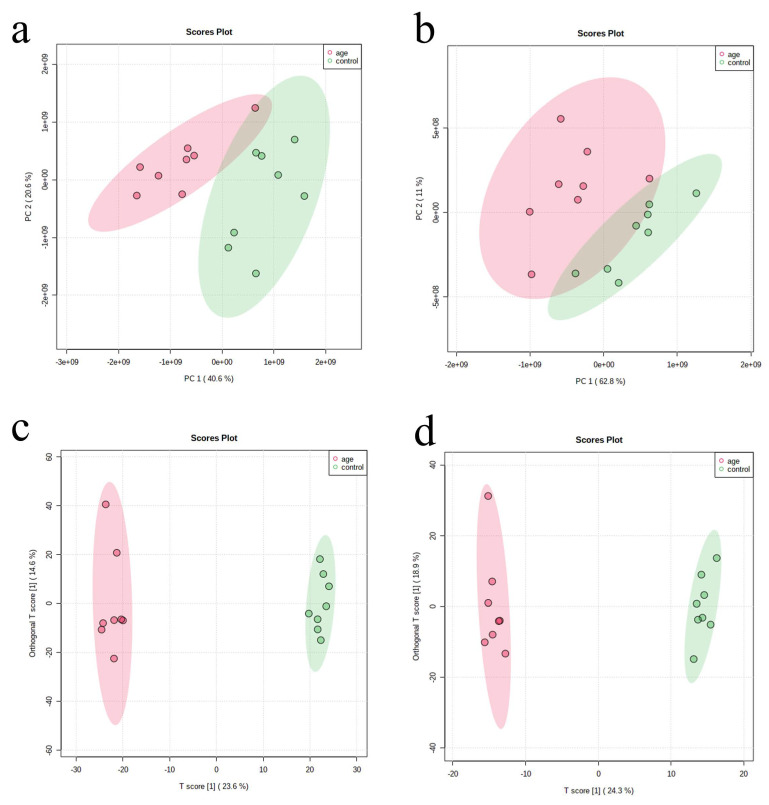
Supervised clustering of serum metabolites using the PCA and OPLS-DA models. (**a**,**b**) Score plot of the PCA model for samples collected from the two isolates of sample data; (**c**,**d**) The two groups were well separated in the OPLS-DA score plot. ((**a**,**c**): under the negative modes; (**b**,**d**): under the positive modes).

**Figure 3 biomolecules-12-01594-f003:**
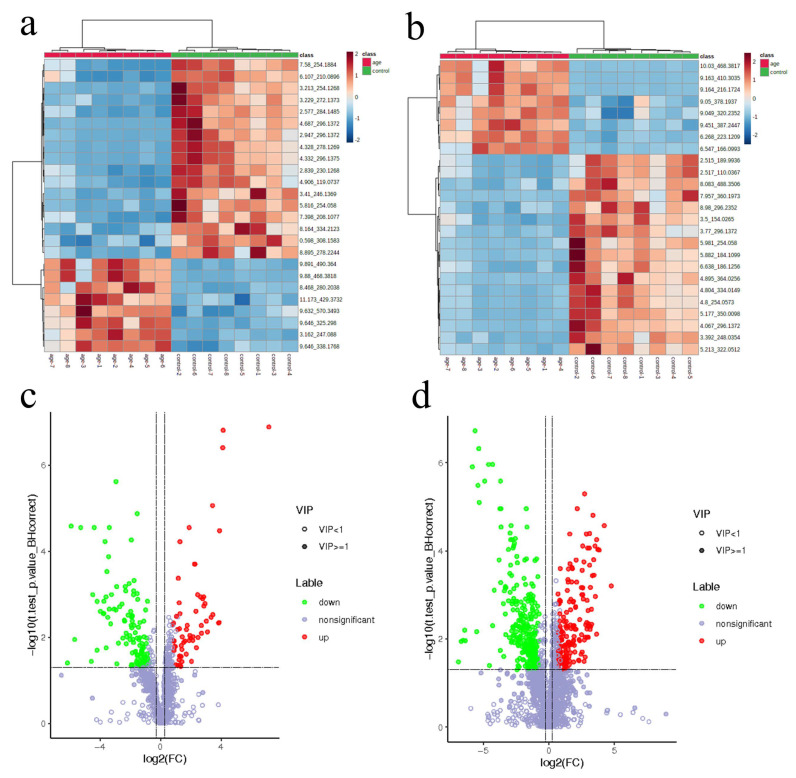
Altered serum metabolomics in the aging group compared with the control. (**a**,**b**) The heatmap of untargeted plasma metabolomics showing the top 20 metabolites that are different between the aging and control group; (**c**,**d**) The volcano plots of the different metabolic characteristics in the two modes. ((**a**,**c**): neg; (**b**,**d**): pos).

**Figure 4 biomolecules-12-01594-f004:**
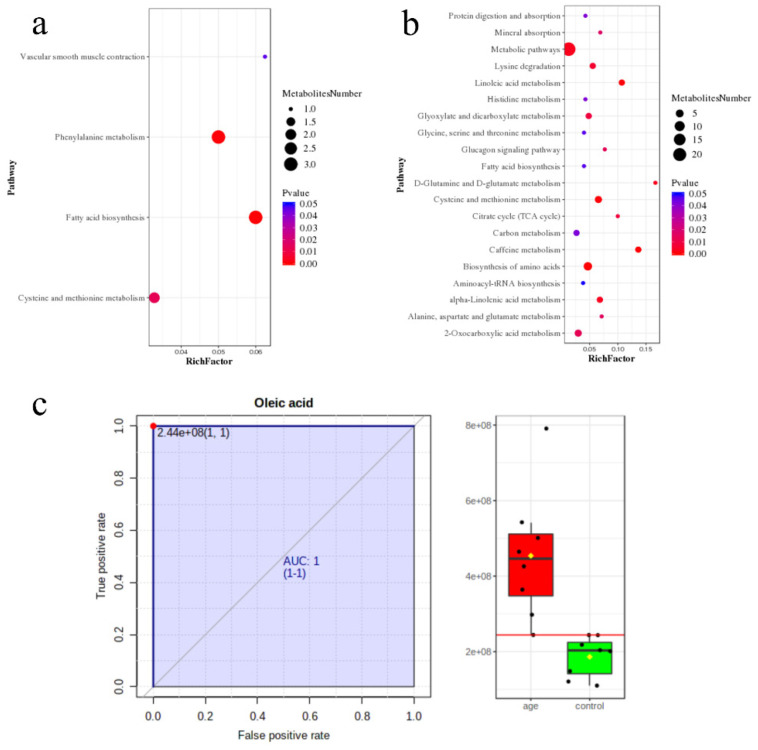
Untargeted metabolomics pathway analysis. (**a**) The pathway enrichment analysis in negative mode. (**b**) The pathway enrichment analysis in positive mode. The color and size of each circle was based on the *p*-value and pathway impact value, respectively. (**c**) The ROC analysis of oleic acid for differentiating the aging group from the control.

**Figure 5 biomolecules-12-01594-f005:**
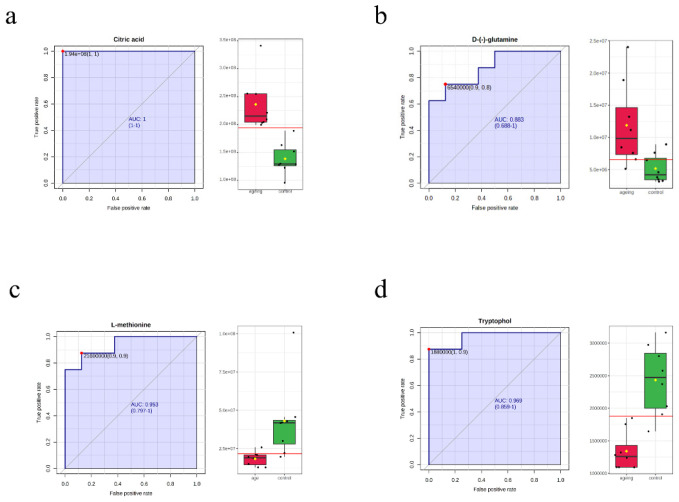
The ROC curve analysis of potential biomarkers (AUC > 0.85) for differentiating the aging group from the control. (**a**) Citric acid, (**b**) D-(-)-glutamine, (**c**) L-methionine, and (**d**) Tryptophol

**Table 1 biomolecules-12-01594-t001:** Candidate markers identified from metabolomic profiling (pos-up: the up-regulated metabolites in positive mode; pos-down: the down-regulated metabolites in positive mode; neg-up: the up-regulated metabolites in negative mode; neg-down: the down-regulated metabolites in negative mode).

Metabolite	Fold Change	*p*-Value	*q*-Value	VIP	Label	Identification Level
Ophthalmic acid	11.6266	0	0.0011	2.3414	pos-up	Level 1
Oleoyl ethanolamide	3.0123	0	0	1.645	pos-up	Level 2
Oleate	1.8467	0.0001	0.0014	1.175	pos-up	Level 2
Citric acid	1.7038	0.0001	0.0016	1.1059	pos-up	Level 4
Alpha-ketoglutaric acid	1.7726	0.0001	0.0013	1.1461	pos-up	Level 4
Linamarin	12.4902	0	0.0001	2.5118	pos-up	Level 4
L-methionine	0.4194	0.0001	0.0012	1.3447	pos-down	Level 1
Pantothenic acid	0.4005	0	0.0006	1.4818	pos-down	Level 1
Formyl-l-methionyl peptide	0.2699	0.0001	0.0018	1.7884	pos-down	Level 1
N-acetyl-l-phenylalanine	0.5542	0	0.0004	1.1529	pos-down	Level 1
Genistein	0.055	0	0.0008	2.7982	pos-down	Level 2
9-oxo-10(e),12(e)-octadecadienoic acid	0.1687	0	0.0001	2.1344	pos-down	Level 2
Gamma-linolenic acid	0.1501	0.0001	0.0022	2.2762	pos-down	Level 2
Tryptophol	0.5508	0.0001	0.0018	1.142	pos-down	Level 4
Palmitoleic acid2	4.7506	0	0.0002	1.7995	neg-up	Level 2
4-hydroxybenzoic acid	0.2634	0	0	0.0001	neg-down	Level 2
Fmet	0.2899	0	0	0.001	neg-down	Level 4
3-phenyllactic acid	0.5047	0.0001	0.0001	0.0023	neg-down	Level 2
Pantothenic acid	0.3525	0.0001	0.0001	0.0013	neg-down	Level 2
Genistein	0.0826	0	0	0.0012	neg-down	Level 2

## Data Availability

The raw metabolomics data from MetaboLights (https://www.ebi.ac.uk/metabolights/, accessed on 8 October 2022) are under study number MTBLS6107. Our code is publicly available at https://github.com/yuetong240028/ML-datamining (accessed on 8 October 2022). The datasets used and analyzed are available from the corresponding author upon reasonable request during the current study.

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
