# Peer review of "Serum Metabolomic Profiling in Aging Mice Using Liquid Chromatography—Mass Spectrometry"

_biomolecules, 2022, doi:10.3390/biom12111594_

Round 1
Reviewer 1 Report
In this manuscript, Yue et al. describe the metabolic profiling in aging mice using LC-MS, where they have found 54 biomarkers between 18-month-old and 3-month-old mice. They hihglight the importance of the biomarker discovery for aging-related diseases. The authors follow a well stablished scientific approach. The manuscript is well written and, as far as I can judge based on the current manuscript, the technical quality of the work is satisfactory. However, the manuscript needs to be improved in a few aspects. There is a key aspect missed that should be mandatory for all experiments. The raw data should be available in a data repository as Metabolomics Workbench or Massive so anyone can replicate the analysis. If so, it would be appreciated a small documentation about the data and its processing. Some software is proprietary, therefore the analysis could be replicated following the methods described in the paper. The use of ML methods to rank the biomarkers is highly appreciated.
Remarks organised per section:
Introduction:
some references are missed in lines 59-61. The authors link 4 disease types with aging, but only 1 is referenced.
Lines 64-67 should be rewritten. The authors present two independent studies that have found aging biomarkers, and they connect both while they are not related.
The sentence in lines 70-71 "Furthermore, the disease mechanisms can differ among individuals, necessitating precision medicine in the future." needs a reference.
Materials and methods:
The units in sample preparation are right now read as L, where it should be microL.
The writing could be improved. There are sentences with no verb (lines 135-136).
Figure 1 shows the workflow used for this analysis. However, the annotation/identification is not shown there. Also, the steps of this workflow should be clearly labelled with sequence number to facilitate the comprehension and reproducibility.
Results:
The confidence level published by MSI of metabolite identification should be specified. It is not stated if MS/MS was used for all metabolites, if authentic standards were available and those identifications were confirmed, or if the authors used just the m/z ratio. This is key in the communication of the results since the dicussion is completely depending on the proper identification of compounds.
Discussion:
This section is very well written and easy to read. The limitations stated here are highly appreciated.
Author Response
Response to Reviewer 1 Comments
In this manuscript, Yue et al. describe the metabolic profiling in aging mice using LC-MS, where they have found 54 biomarkers between 18-month-old and 3-month-old mice. They hihglight the importance of the biomarker discovery for aging-related diseases. The authors follow a well stablished scientific approach. The manuscript is well written and, as far as I can judge based on the current manuscript, the technical quality of the work is satisfactory. However, the manuscript needs to be improved in a few aspects. There is a key aspect missed that should be mandatory for all experiments. The raw data should be available in a data repository as Metabolomics Workbench or Massive so anyone can replicate the analysis. If so, it would be appreciated a small documentation about the data and its processing. Some software is proprietary, therefore the analysis could be replicated following the methods described in the paper. The use of ML methods to rank the biomarkers is highly appreciated.
Author response: Thank you for taking the time to review our manuscript. We appreciate your recognition of our work and valuable comments. The raw metabolomics data from MetaboLights (https://www.ebi.ac.uk/metabolights/) under study number MTBLS6107 and will be released after the article is published. Our code is publicly available at https://github.com/yuetong240028/ML-datamining.
Remarks organised per section:
Introduction:
some references are missed in lines 59-61. The authors link 4 disease types with aging, but only 1 is referenced.
Author response: Thank you for pointing this out, adding related references can improve the quality of the introduction section. We added the references [11,12]: The Emerging Role of Metabolomics in the Diagnosis and Prognosis of Cardiovascular Disease. J Am Coll Cardiol 2016, 68, 2850-2870 and Non-targeted metabolomic biomarkers and metabotypes of type 2 diabetes: A cross-sectional study of PREDIMED trial participants. Diabetes Metab 2019, 45, 167-174 in Lines 63-64.
Lines 64-67 should be rewritten. The authors present two independent studies that have found aging biomarkers, and they connect both while they are not related.
Author response: Thank you for pointing this out. We have corrected the sentence in Lines 66-71. For instance, 14 circulating biomarkers were independently related to all-cause mortality in a recent analysis of large populations [13]. The metabolomics analysis in blood of 15 young and 15 elderly people showed that blood metabolites is highly valuable for human aging research. These metabolites were considered promising and convenient approaches to monitoring aging and related diseases.
The sentence in lines 70-71 "Furthermore, the disease mechanisms can differ among individuals, necessitating precision medicine in the future." needs a reference.
Author response: Thank you for pointing this out. We have added the reference [15]: Cell. 2020 Apr 2;181(1):92-101 in Lines 74-75. The article referred machine learning hold promises for future rigorous, outcomes-based medicine with detection, diagnosis, and treatment strategies that are continuously adapted to individual and environmental differences.
Materials and methods:
The units in sample preparation are right now read as L, where it should be microL.
Author response: Thank you for pointing this out. We have recorrected the units in Lines 98-99. Internal standards mix 1 (IS1) and internal standards mix 2 (IS2) were added to 100 μL samples. And samples extracted by immediately adding 300 L of precooled methanol and acetonitrile (2:1, v/v) to 100 μL samples for quality control of sample preparation.
The writing could be improved. There are sentences with no verb (lines 135-136).
Author response: Thank you for pointing this out. We have recorrected the sentence: Four algorithms, Logistic Regression, Support Vector Machine, Gaussian Naive Bayes, and Random Forest, were used for machine learning by using Python 3 in Lines 139-141.
Figure 1 shows the workflow used for this analysis. However, the annotation/identification is not shown there. Also, the steps of this workflow should be clearly labelled with sequence number to facilitate the comprehension and reproducibility.
Author response: Thank you for your suggestion. We have added the annotation and label to facilitate the comprehension and reproducibility. a: Serum samples were collected from aging mice and control mice, and the metabolites data were obtained by LC-MS/MS. b-c: Data processing by peak extraction, peak alignment, and metabolites identification. d: The results of Fold change and Student's t test obtained from univariate analysis were combined to screen for differential metabolites. e: PCA and OPLS-DA were used to reduce the dimensionality of multivariate raw data, and the similarity and difference between and within the sample groups are obtained. f-g: Metabolite classification and functional annotation were carried out, and machine learning was used to search for biomarkers.
Results:
The confidence level published by MSI of metabolite identification should be specified. It is not stated if MS/MS was used for all metabolites, if authentic standards were available and those identifications were confirmed, or if the authors used just the m/z ratio. This is key in the communication of the results since the dicussion is completely depending on the proper identification of compounds.
Author response: Thank you for pointing this out. According to the information that can be used for matching (including MS1 molecular weight, MS2 fragment ion spectrum, column retention time, presence or absence of reference standards, etc.), the identified substances are annotated with a level of confidence, which is divided into different credibility levels. Level 1: Result from BGI standard library, and the compound is with MS2 spectrum score greater than or equal to 60 and the RT deviation within 0.2 min. Level 2: Result from BGI standard library or mzCloud standard library and the compound is without RT value or RT deviation greater than 0.2min. Level 3: Result from BGI standard library or mzCloud standard library, but the MS2 spectrum score is less than 60. Level 4: Result from ChemSpider. Level 5: Exact molecular weight (MW). There is no match in the database, that is, no identification results. Among them, the reliability of Level 1 to Level 5 decreases in turn. We examined the metabolites mentioned in the results. Most of the compounds were Level 1or Level 1, and only a few were Level 4.
Discussion:
This section is very well written and easy to read. The limitations stated here are highly appreciated.
Author response: Special thanks to you for your favorable comments.
Reviewer 2 Report
The manuscript presents a metabolomics analysis of aging mouse serum vs non-aged controls. The experimental design includes an n=8 per group, which is adequate for standard biostatistical analysis approaches, but is typically inadequate for most machine learning approaches that require larger sample sizes. The biostatistical analyses are reasonably described for the most part and appear rigorous. The machine learning analyses are woefully under-described. Also, the results are superficially interpreted with respect to mammalian metabolism. The bacterial metabolic pathway D-glutamine and glutamate metabolism is highlighted, which requires monumental justification, considering the context is mammalian metabolism and not bacterial metabolism. Also, the isoflavone metabolite genistein is indicated in the differential results. But this metabolite could only come from diet and the manuscript includes no description of the mouse diet.
Major Issues:
1. Line 31: change “54 metabolites differed compare with control mice” to “54 metabolites were differential when compared to control mice at an adjusted p-value < XXX”. This provides information needed to evaluate the statistical significance of the results. Could add other filtering criteria used to distill down to these 54 metabolites.
2. The association of oleic acid with aging has been known for a long time. Therefore, this is not that novel. See the following reference: Hollander, Daniel, and Violetta D. Dadufalza. "Increased intestinal absorption of oleic acid with aging in the rat." Experimental gerontology 18.4 (1983): 287-292. https://doi.org/10.1016/0531-5565(83)90039-6 . However, this now begs the question of whether oleic acid was in the diet of the mice. But there is no description of the mouse diet. This would fly into the face of the authors’ interpretation that fatty acid biosynthesis is higher in aging mice if the diet is simply providing the fatty acids.
3. D-Glutamine and D-glutamate metabolism occurs in bacteria. So the linkage to mouse serum levels requires context and interpretation, otherwise most of the KEGG pathway analyses become highly suspect. The only known mammalian enzyme involving D-glutamine was just recently discovered, but represents D-glutamine catabolism: Ariyoshi, Makoto, et al. "D-Glutamate is metabolized in the heart mitochondria." Scientific reports 7.1 (2017): 1-9. https://doi.org/10.1038/srep43911
4. Line 138: Need to add more detail on PCA and OPLS-DA analyses into the main text and the Figure 2 legend. Figure 2 indicates “two isolates of sample data” were separately analyzed, but no details of this is in the methods. It is implied in the Results section, that these two separate datasets are from positive and negative ion mode spectra. But this is not explicitly described and the Figure 2 legend definitely does not have this required level of detail. What is in the parentheses “(ac: neg; bd: pos)” is not enough explanation. Also, the PCA results description in the main text should have more detail of the detected separation and the amount of variance in the first 2 PCs.
5. Line 154: Given that q-value (adjusted p-value) was an initial filtering criteria, why wasn’t the stricter filtering criteria using q-value, like q-value < 0.001? This would leave out any question of multiple testing issues in the filter criteria.
6. Table 1: Need an explanation for what pos-up, pos-down, neg-up, and neg-down mean. I interpret this to mean positive vs negative ion mode and whether the metabolite was up-changed or down-changed. But this should be made explicit in a table foot note and/or the main text.
7. Genistein is an isoflavone that mostly comes from soy products. So did this come from the diet? There is no description of the diet for the mice. The authors must verify that the diet contained a likely source for genistein.
8. Figure 5 legend does not match the graphs in the figure. This must be fixed.
9. Line 304: the four machine learning algorithms should be spelled out and their acronym associated with their full name. It is also good form to add references to these methods.
10. There is not description of the machine learning methods in the Methods section. There is no description of training and test sets, especially for generating AUCs. It is hard to see how a dataset of 8 controls and 8 cases could have both adequate training and test sets that could generate meaningful AUCs. Also, it appears that feature selection is the main purpose for using the machine learning methods. But this is not explicitly stated.
Minor Issues:
Line 26: change “However, only a few research have…” to “However, little research has…”
Line 29: change “Machine learning were” to “Machine learning was”
Author Response
Response to Reviewer 2 Comments
The manuscript presents a metabolomics analysis of aging mouse serum vs non-aged controls. The experimental design includes an n=8 per group, which is adequate for standard biostatistical analysis approaches, but is typically inadequate for most machine learning approaches that require larger sample sizes. The biostatistical analyses are reasonably described for the most part and appear rigorous. The machine learning analyses are woefully under-described. Also, the results are superficially interpreted with respect to mammalian metabolism. The bacterial metabolic pathway D-glutamine and glutamate metabolism is highlighted, which requires monumental justification, considering the context is mammalian metabolism and not bacterial metabolism. Also, the isoflavone metabolite genistein is indicated in the differential results. But this metabolite could only come from diet and the manuscript includes no description of the mouse diet.
Author response: Thank you for your constructive comments on our manuscript. We appreciate your recognition of our work and valuable comments. Machine learning requires a large sample size, so we use different algorithms to look for potential markers and support vector machine is suitable for small samples and we recognize this is a study limitation (which was specified in line 410-412. We provide the description of the mouse diet (supplemental table S3) and modify the discussion section to improve our paper. We have studied comments carefully and have made the correction which we hope meet with approval.
Major Issues:
- Line 31: change “54 metabolites differed compare with control mice” to “54 metabolites were differential when compared to control mice at an adjusted p-value < XXX”. This provides information needed to evaluate the statistical significance of the results. Could add other filtering criteria used to distill down to these 54 metabolites.
Author response: Thank you for pointing this out. We have re-written this sentence in Lines 31-32. In total, aging mice are characterized by 54 metabolites were differential when compared to control mice at a criteria: VIP≥1, q-value<0.05, Fold-Change≥1.2 or ≤0.83.
- The association of oleic acid with aging has been known for a long time. Therefore, this is not that novel. See the following reference: Hollander, Daniel, and Violetta D. Dadufalza. "Increased intestinal absorption of oleic acid with aging in the rat." Experimental gerontology 18.4 (1983): 287-292. https://doi.org/10.1016/0531-5565(83)90039-6 . However, this now begs the question of whether oleic acid was in the diet of the mice. But there is no description of the mouse diet. This would fly into the face of the authors’ interpretation that fatty acid biosynthesis is higher in aging mice if the diet is simply providing the fatty acids.
Author response: Thank you for pointing this out. We have also reviewed and discussed oleic acid and aging research about alterations in oleic acid and lipid metabolism have been implicated in various metabolic diseases and aging. As you referred the reference, they observed increase in absorption of oleic acid with aging. This is a very good complement to our results, which we have also discussed in Lines 365-366. A study has shown that as the rats aged, the absorption of oleic acid in the intestine of rats increased [34]. And the aging mice and control mice were fed a chow diet to eliminate confounders. The caloric composition of the chow diet was 74.9% carbohydrate, 4.8% fat, and 20.3% protein (TableS3).
- D-Glutamine and D-glutamate metabolism occurs in bacteria. So the linkage to mouse serum levels requires context and interpretation, otherwise most of the KEGG pathway analyses become highly suspect. The only known mammalian enzyme involving D-glutamine was just recently discovered, but represents D-glutamine catabolism: Ariyoshi, Makoto, et al. "D-Glutamate is metabolized in the heart mitochondria." Scientific reports 7.1 (2017): 1-9. https://doi.org/10.1038/srep43911
Author response: Thank you for pointing this out. Your comments enhance the rigor of our article. We have revised discussion to include mentioned reference and other glutamate-related literature in Lines 395-401. Glutamine is the most common amino acid in circulation, and it has been linked to the TCA cycle [47]. D-glutamic acid was first detected in mammalian tissues, and glutamine metabolism is associated with advanced age [48,49]. A study has shown that different metabolites present in 3-month-old and 24-month-old male mice are involved in D-glutamine and D-glutamate metabolism, which may contribute to changes in renal dysfunction [50]. In addition, age-related disorders such as major depressive disorder (MDD), individuals with MDD had a significantly higher ratio of glutamine to glutamate than normal older adults at baseline [51].
- Line 138: Need to add more detail on PCA and OPLS-DA analyses into the main text and the Figure 2 legend. Figure 2 indicates “two isolates of sample data” were separately analyzed, but no details of this is in the methods. It is implied in the Results section, that these two separate datasets are from positive and negative ion mode spectra. But this is not explicitly described and the Figure 2 legend definitely does not have this required level of detail. What is in the parentheses “(ac: neg; bd: pos)” is not enough explanation. Also, the PCA results description in the main text should have more detail of the detected separation and the amount of variance in the first 2 PCs.
Author response: Thank you for pointing this out. We used a multivariate data analysis including the Pareto-scaled principal component analysis (PCA) and orthogonal partial least-squares discriminant analysis (OPLS-DA) to evaluate the model validity for the serum samples. PCA is an unsupervised method aiming to find the directions that best explain the variance in a dataset (X) without referring to class labels (Y). The data are summarized into much fewer variables called scores which are weighted average of the original variables. The weighting profiles are called loadings. OPLS-DA, like PLS-DA, is a powerful tool used for dimension reduction and identification of spectral features that drive group separation. We have added the detail of the detected separation and the amount of variance of PCA in Lines 153-155. In addition, we also found our errors in the article writing process and corrected them.
- Line 154: Given that q-value (adjusted p-value) was an initial filtering criteria, why wasn’t the stricter filtering criteria using q-value, like q-value < 0.001? This would leave out any question of multiple testing issues in the filter criteria.
Author response: Thank you for pointing this out. Here we used p < 0.0001 and q-value < 0.01 to further select the altered metabolites, and the initial standard choice is q-value < 0.05. We have added q-value < 0.01 in Line 167.
- Table 1: Need an explanation for what pos-up, pos-down, neg-up, and neg-down mean. I interpret this to mean positive vs negative ion mode and whether the metabolite was up-changed or down-changed. But this should be made explicit in a table foot note and/or the main text.
Author response: Thank you for pointing this out. We have added the Table foot note in the legend and main text in Lines 199-202.
- Genistein is an isoflavone that mostly comes from soy products. So did this come from the diet? There is no description of the diet for the mice. The authors must verify that the diet contained a likely source for genistein.
Author response: Thank you for pointing this out. We have soybean oil in mice diet ingredients, and as you mentioned, other studies have mentioned that soybean meal products produce genistein. https://doi.org/10.1093/cdn/nzaa031. We have added the diet information in Lines 88-90 and supplemental table S3.
- Figure 5 legend does not match the graphs in the figure. This must be fixed.
Author response: Thank you for pointing this out. We are very sorry for our negligence of the figure 5 and we have made the corrected figure match the legend.
- Line 304: the four machine learning algorithms should be spelled out and their acronym associated with their full name. It is also good form to add references to these methods.
Author response: Thank you for pointing this out. We have added the four machine learning algorithms acronym associated with their full name in Lines 328-334 to facilitate the comprehension.
- There is not description of the machine learning methods in the Methods section. There is no description of training and test sets, especially for generating AUCs. It is hard to see how a dataset of 8 controls and 8 cases could have both adequate training and test sets that could generate meaningful AUCs. Also, it appears that feature selection is the main purpose for using the machine learning methods. But this is not explicitly stated.
Author response: Thank you for pointing this out. All four algorithms of machine learning are 8-cross validation using the following code: KF = Kfold (N, shuffle = True). And we added into the methods: The four algorithms of machine learning use 8-cross validation to randomly divide the data set into training data and test data in Lines 145-146. ROC curves are generated by Monte-Carlo cross validation (MCCV) using balanced sub-sampling. In each MCCV, two thirds (2/3) of the samples are used to evaluate the feature importance. The top 2, 3, 5, 10 ...100 (max) important features are then used to build classification models which is validated on the 1/3 the samples that were left out. The procedure was repeated multiple times to calculate the performance and confidence interval of each model. We hope this description will explain our approach to machine learning.
Minor Issues:
Line 26: change “However, only a few research have…” to “However, little research has…”
Author response: Thank you for pointing this out. We have re-written this sentence in Lines 26-27. However, little research has found meaningful markers that reflect aging profiles based on machine learning.
Line 29: change “Machine learning were” to “Machine learning was”
Author response: Thank you for pointing this out. We have changed this word in Line 30. Machine learning was used to screen aging-related biomarkers.
Round 2
Reviewer 1 Report
After the first review, Yue et al. have adressed most of the concerns of the reviewers. The general description, the abstract and the introduction have been improved. However, taking into account that the sample size is limited for ML approaches, it should be referenced some background of ML approaches with limited sample sizes. The authors made a great effort publishing the experimental data in Metabolights and addressing the issues in the introduction, but an explanation and a justification of the use of ML will improve the article.
In the methods section Figure 1, the authors updated the Figure 1 description, but not the figure itself. The annotation/identification process should be incorporated to the step c as it is explained in the Figure description.
In the same Methods section, he authors explain how the MCCV was performed to the reviewer 2:
"In each MCCV, two thirds (2/3) of the samples are used to evaluate the feature importance. The top 2, 3, 5, 10 ...100 (max) important features are then used to build classification models which is validated on the 1/3 the samples that were left out. The procedure was repeated multiple times to calculate the performance and confidence interval of each model. We hope this description will explain our approach to machine learning." This should be stated in the manuscript.
Analogously to this explanation, the authors explain to the reviewer 1 the confidence level for their annotations:
"Level 1: Result from BGI standard library, and the compound is with MS2 spectrum score greater than or equal to 60 and the RT deviation within 0.2 min. Level 2: Result from BGI standard library or mzCloud standard library and the compound is without RT value or RT deviation greater than 0.2min. Level 3: Result from BGI standard library or mzCloud standard library, but the MS2 spectrum score is less than 60. Level 4: Result from ChemSpider. Level 5: Exact molecular weight (MW). There is no match in the database, that is, no identification results. Among them, the reliability of Level 1 to Level 5 decreases in turn. We examined the metabolites mentioned in the results. Most of the compounds were Level 1or Level 1, and only a few were Level 4."
However, each biomarker identification should be included in the table S2 (Serum biomarkers). This information is key to provide confidence in the reader about the pathway analysis and the biological conclusions. Level 4 annotations cannot be handled with the same confidence than level 1 identifications. The explanation about the identification process should be introduced in the manuscript. In the current form, the manuscript does not provide information about the use of reference standards (BGI standard library) to perform level 1 identifications.
The manuscript has improved and it is very well explained, but these two concerns should be addressed prior to publication.
Author Response
Response to Reviewer 1 Comments
After the first review, Yue et al. have adressed most of the concerns of the reviewers. The general description, the abstract and the introduction have been improved. However, taking into account that the sample size is limited for ML approaches, it should be referenced some background of ML approaches with limited sample sizes. The authors made a great effort publishing the experimental data in Metabolights and addressing the issues in the introduction, but an explanation and a justification of the use of ML will improve the article.
Author response: Thank you very much for reviewing our manuscript again. We appreciate your recognition of our work and valuable comments. We have carried out the explanation and a justification of the use of ML that the reviewers suggested and revised the manuscript accordingly. We hope that you find our responses satisfactory.
In the methods section Figure 1, the authors updated the Figure 1 description, but not the figure itself. The annotation/identification process should be incorporated to the step c as it is explained in the Figure description.
Author response: Thank you for pointing this out. We have labeled the figure1 in Lines 181-193.
In the same Methods section, he authors explain how the MCCV was performed to the reviewer 2:
"In each MCCV, two thirds (2/3) of the samples are used to evaluate the feature importance. The top 2, 3, 5, 10 ...100 (max) important features are then used to build classification models which is validated on the 1/3 the samples that were left out. The procedure was repeated multiple times to calculate the performance and confidence interval of each model. We hope this description will explain our approach to machine learning." This should be stated in the manuscript.
Author response: Thank you for pointing this out. We agree with the reviewer that further elaborating on this point using the ML method would be helpful. We have added the sentence: “Receiver operator characteristic (ROC) curve analysis was generated by Monte-Carlo cross validation (MCCV) using balanced sub-sampling. “in Lines 149-153.
Analogously to this explanation, the authors explain to the reviewer 1 the confidence level for their annotations: "Level 1: Result from BGI standard library, and the compound is with MS2 spectrum score greater than or equal to 60 and the RT deviation within 0.2 min. Level 2: Result from BGI standard library or mzCloud standard library and the compound is without RT value or RT deviation greater than 0.2min. Level 3: Result from BGI standard library or mzCloud standard library, but the MS2 spectrum score is less than 60. Level 4: Result from ChemSpider. Level 5: Exact molecular weight (MW). There is no match in the database, that is, no identification results. Among them, the reliability of Level 1 to Level 5 decreases in turn. We examined the metabolites mentioned in the results. Most of the compounds were Level 1or Level 1, and only a few were Level 4." However, each biomarker identification should be included in the table S2 (Serum biomarkers). This information is key to provide confidence in the reader about the pathway analysis and the biological conclusions. Level 4 annotations cannot be handled with the same confidence than level 1 identifications. The explanation about the identification process should be introduced in the manuscript. In the current form, the manuscript does not provide information about the use of reference standards (BGI standard library) to perform level 1 identifications.
Author response: Thank you for pointing this out. We agree with the reviewer that further elaborating on this point using the identification process would be helpful. First, we have added the confidence level of candidate markers in Table1. And we have added the confidence level of serum biomarkers by machine learning in the Table S2. We added the following sentence to Lines 117-118 in the methods: “Identification of metabolites meet the levels published by the Metabolomics Standards Initiative (MSI)”. Level 1 identifications represent the ideal situation, where the proposed structure has been confirmed via appropriate measurement of a reference standard with MS, MS/MS and retention time matching.
The manuscript has improved and it is very well explained, but these two concerns should be addressed prior to publication.
Author response: Once again, thank you very much for your constructive comments and suggestions which would improve the quality of our manuscript.
Reviewer 2 Report
The authors have addressed many of my issues.
However one major issue still remains. There is no adequate explanation and justification for D-glutamine and D-glutamate metabolism in mammalian metabolism.
Jiao et al., 2022 mentions detecting the same pathway, but their justification is very weak. They cite the same paper you cite, Meynial-Denis 2016, which never mentions D-glutamine. Also, they used pathway enrichment analysis of KEGG pathways. KEGG pathways are NOT limited to mammalian metabolism and are very E-coli centric.
D-glutamine has been detected in mammalian tissues, but mammalian metabolism does not do anything with it. Here are two references about D-glutamine in mammals:
Meister, Alton. "23. Glutamine synthetase of mammals." The enzymes. Vol. 10. Academic Press, 1974. 699-754.
Renick, Paul J., et al. "Imaging of actively proliferating bacterial infections by targeting the bacterial metabolic footprint with d-[5-11C]-glutamine." ACS infectious diseases 7.2 (2021): 347-361.
The first is a classic reference and indicates that D-glutamine has been detected at low levels in mammalian tissue. The second reference provides an explanation for its detection in mammals as being from bacterial sources including infection and that it is cleared very efficiently by the kidneys.
You used MetaboAnalyst 5.0 for the pathway enrichment analysis, but did you limit to human metabolism?
A second minor issue is that you mentioned using pareto-scaled PCA. You need to add this detail into your methods.
Author Response
Response to Reviewer 2 Comments
The authors have addressed many of my issues.
Author response: Thank you for reading our manuscript and reviewing it again, which will help us improve the quality of this manuscript.
However one major issue still remains. There is no adequate explanation and justification for D-glutamine and D-glutamate metabolism in mammalian metabolism.
Author response:
We sincerely appreciate the valuable comments.
At first, we didn't notice that this pathway only occurs in E. coli. We queried the D-Glutamine and D-Glutamate Metabolism in the PubChem (https://pubchem.ncbi.nlm.nih.gov/pathway/PathBank:SMP0000792), and its Taxonomy is Escherichia coli.
We think it is very meaningful for you to raise the question. We looked at other papers and found that a lot of the metabolomics papers showed changes in D-glutamine and D-glutamate metabolism pathway. We think that because D-Glutamine changes in aging mice in our study, and D-Glutamine is involved in the D-glutamine and D-glutamate metabolism, so this pathway is enriched. We found that the D-glutamine and D-glutamate metabolism pathway is enriched in mice and humans because of changes in D-Glutamine in other metabolomics articles such as:
Zhou D et al., Integrated Metabolomics and Proteomics Analysis of Urine in a Mouse Model of Posttraumatic Stress Disorder. Front Neurosci. 2022 Mar 11;16:828382. doi: 10.3389/fnins.2022.828382.
Patel D et al., Alterations in bone metabolites with age in C57BL/6 mice model. Biogerontology. 2022 Oct;23(5):629-640. doi: 10.1007/s10522-022-09986-7.
Xiao L et al., Nontargeted Metabolomic Analysis of Plasma Metabolite Changes in Patients with Adolescent Idiopathic Scoliosis. Mediators Inflamm. 2021 May 25;2021:5537811. doi: 10.1155/2021/5537811.
Jia R et al., Characteristics of serum metabolites in sporadic amyotrophic lateral sclerosis patients based on gas chromatography-mass spectrometry. Sci Rep. 2021 Oct 21;11(1):20786. doi: 10.1038/s41598-021-00312-8.
Lu C et al.,Comprehensive metabolomic characterization of atrial fibrillation. Front Cardiovasc Med. 2022 Aug 8;9:911845. doi: 10.3389/fcvm.2022.911845.
Song J et al., The anti-aging effect of Scutellaria baicalensis Georgi flowers extract by regulating the glutamine-glutamate metabolic pathway in d-galactose induced aging rats. Exp Gerontol. 2020 Feb 8;134:110843. doi: 10.1016/j.exger.2020.110843.
Especially in this article (Song J et al.), they mentioned “In this study, pathway impact value > 0.10 was regarded as target metabolic pathways related to aging included (1) D-glutamine and D-glutamate metabolism. These findings indicated the greatest pathway impact of glutamine and glutamate metabolism may be an important factor in process of aging..” But what they don't mention in the discussion is the role of D-glutamine and the D-glutamine and D-glutamate metabolism pathways involved.
Therefore, in the revised manuscript, we discussed further as below to give adequate explanation and justification for D-glutamine and D-glutamate metabolism in mammalian metabolism: However, D-Glutamine and D-glutamate metabolism occurs in bacteria [53]. It is plausible that altered gut microbiota composition, the altered percentage of D-Glutamine and D-glutamate-producing bacteria in particular, may lead to deregulated metabolism of D-Glutamine and D-glutamate in aging mouse serum. Further suggestion in this clarifying this issue from the reviewer is highly appreciated.
Jiao et al., 2022 mentions detecting the same pathway, but their justification is very weak. They cite the same paper you cite, Meynial-Denis 2016, which never mentions D-glutamine. Also, they used pathway enrichment analysis of KEGG pathways. KEGG pathways are NOT limited to mammalian metabolism and are very E-coli centric.
Author response: Thank you for pointing this out. Meynial-Denis 2016, the article you mentioned is indeed full of Glutamine metabolism in advanced age, not to mention D-glutamine. As we mentioned before, they also found that D-Glutamine changes and it is involved in the D-glutamine and D-glutamate metabolism pathway. Please refer to above answer for adequate explanation and justification for D-glutamine and D-glutamate metabolism in mammalian metabolism.
D-glutamine has been detected in mammalian tissues, but mammalian metabolism does not do anything with it. Here are two references about D-glutamine in mammals:
Meister, Alton. "23. Glutamine synthetase of mammals." The enzymes. Vol. 10. Academic Press, 1974. 699-754.
Renick, Paul J., et al. "Imaging of actively proliferating bacterial infections by targeting the bacterial metabolic footprint with d-[5-11C]-glutamine." ACS infectious diseases 7.2 (2021): 347-361.
The first is a classic reference and indicates that D-glutamine has been detected at low levels in mammalian tissue. The second reference provides an explanation for its detection in mammals as being from bacterial sources including infection and that it is cleared very efficiently by the kidneys.
Author response: We also read and learned about the two articles you mentioned. We agree that D-glutamine has been detected at low levels in mammalian tissue and cleared very efficiently by the kidneys.
“Han M., et al. Development and validation of a rapid, selective, and sensitive LC-MS/MS method for simultaneous determination of D- and L-amino acids in human serum: application to the study of hepatocellular carcinoma. Anal Bioanal Chem. 2018 Apr;410(10):2517-2531. doi: 10.1007/s00216-018-0883-3.”
They mentioned: “The serum concentrations of D-glutamate and D-glutamine were significantly reduced in the hepatocellular carcinoma patients compared with the healthy individuals (P < 0.01). D-Glutamate and D-glutamine were identified as the most downregulated serum markers (fold change greater than 1.5), which deserves further attention in hepatocellular carcinoma research.”
Although these two metabolites are at low levels, they can be detected in human serum by metabolomics methods.
Further suggestion in this clarifying this issue from the reviewer is highly appreciated.
You used MetaboAnalyst 5.0 for the pathway enrichment analysis, but did you limit to human metabolism?
Author response: Thank you for pointing this out. When we used the parameter settings of MetaboAnalyst 5.0 for pathway analysis, we selected a pathway library (KEGG pathway info were obtained in Oct. 2019) limit to Homo sapiens (KEGG).
In our analysis, D-Glutamate and D-glutamine metabolism pathway appears on the KEGG (00471). But the KEGG site recently integrated 00471 into the 00470 pathway, and 00470(KEGG PATHWAY: D-Amino acid metabolism - Reference pathway (genome.jp)) is available, so 00471 is normal after we limited to Homo sapiens.
A second minor issue is that you mentioned using pareto-scaled PCA. You need to add this detail into your methods.
Author response: Thank you for pointing this out. The scaling method used in PCA analysis in this report is a common pareto correction. We have added the sentence: “The metabolites and profile differences between aging and control mice groups were determined using unsupervised principal component analysis (PCA) applied to common pareto correction and supervised projections to latent structures-discriminant analysis (OPLS-DA). “in Lines 138-139.
We would like to thank again the reviewer for careful and thorough reading of this manuscript and for the thoughtful comments and constructive suggestions, which help to improve the quality of this manuscript.